# Toward a Plausible Methodology to Assess Rock Slope Instabilities at a Regional Scale

**Dimitris Sotiriadis** [1,*], **Nikolaos Klimis** [1], **Elisavet Isavela Koutsoupaki** [1], **Eleni Petala** [1], **Sotiris Valkaniotis** [1], **Maria Taftsoglou** [1], **Vasileios Margaris** [2] and **Ioannis Dokas** [1]

1   Department of Civil Engineering, Democritus University of Thrace, University Campus, 67100 Xanthi, Greece
2   Institute of Engineering Seismology and Earthquake Engineering, EPPO-ITSAK, 55555 Thessaloniki, Greece
*   Correspondence: dsotiria@civil.duth.gr

**Abstract:** Slope failures along road cuts and highways, occurring due to heavy rainfalls or earthquakes, pose significant threats to people, vehicles, and emergency plans. In the present study, a methodology to assess the stability of rock slopes at a regional scale is proposed using a kinematic analysis and a probabilistic limit equilibrium analysis for plane sliding and wedge failure modes. The workflow adopted is described through its implementation along the main road network of the island of Thasos, located in northern Greece. On-site investigations and measurements along the island's road network formed the basis of the present study. The results of the kinematic analysis showed that the joint sets, which were identified during the on-site investigations, formed critical intersections that could lead to wedge and plane sliding failures. The on-site measurements and the results of the kinematic analysis were utilized to perform limit equilibrium back-analyses at sites of identified failures due to the water pressure effects to probabilistically estimate the material strength properties of the joints. Subsequently, numerous limit equilibrium analyses were executed within a Monte Carlo simulation framework to produce representative fragility curves of rock slopes against plane sliding and wedge failures along the main road network, due to earthquake loading and water pressures.

**Keywords:** rock slope stability; kinematic analysis; limit equilibrium analysis; fragility curves

## 1. Introduction

Rock slope failures are included in the generic term of geohazards, which are triggered by natural phenomena, such as the development of adverse water pressures, which is often due to heavy rainfalls or/and earthquakes. In mountainous road networks, frequent rockfalls are recorded on road cut-slopes, which occasionally cause critical effects on the smooth operation of the network and the surrounding structures. More specifically, the potential consequences of rock slope failures along the road networks may be characterized as direct, causing structural damage on its structural elements, or indirect, causing transport delays that lead to wider economic, social, and environmental impacts. Therefore, the risk assessment of geo-structures along the main road networks is of the upmost importance. In the literature, various methodologies have been proposed for the seismic risk assessment of road networks [1–4], toward a reduction in losses and unfavorable effects.

Many research efforts have been published considering the stability of rock slopes, either to develop new methodologies, or to investigate the cause of occurred failures. Sepúlveda et al. (2005) [5] conducted field investigations and slope stability back-analyses to confirm the impact of topographic amplification to the triggering of the observed rock sliding and falling failures during the 1994 Pacoima Canyon earthquake. Their analyses included both deterministic and probabilistic pseudo-static approaches, as well as Newmark-type stability analyses [6]. Kveldsvik et al. (2009) [7] utilized the distinct element code UDEC to assess the seismic stability of a very high rock slope, located in western Norway, under seismic scenarios with different return periods. Sumi et al. (2009) [8] proposed a simple method for evaluating the fragility of rock slopes, through limit equilibrium

theory, taking into account both pore-water pressures and seismic forces. Shukla et al. (2009) [9] derived analytical expressions for the factor of safety of rock slopes, incorporating pore-water pressures and seismic forces, as well as the external stabilizing forces provided by anchoring systems. Their goal was to perform an exhaustive parametric study on the effect of surcharge on the stability of rock slopes. Li et al. (2018) [10] carried out a three-dimensional seismic displacement analysis of rock slopes by virtue of the kinematic theorem of the limit analysis and Newmark's method. As part of their work, they calculated the yield coefficient of rock slopes as a function of the material strength and slope geometry parameters. Newmark's method was modified by Cui et al. (2019) [11], to consider the dynamic evolution of the yield acceleration of rock slopes through the Monte Carlo simulation. An improved version of the Newmark analysis was also presented by Zang et al. (2020) [12] for mapping hazards of co-seismic landslides. In many efforts found in the literature, the inherent variability of rock mass properties has been considered through a probabilistic, stochastic, or reliability analysis of the stability of rock slopes. Among others, Wu et al. (2015) [13] presented a methodology for constructing fragility functions to characterize slope instability under a range of damaging earthquakes and rainfalls, including the First Order Reliability Method (FORM) and the copula-based sampling method (CBSM). Furthermore, Roy et al. (2018) [14] considered the variability in geological properties on the slope stability within the framework of FORM, using the Response Surface Method (RSM), to identify the critical design parameters along with the quantification of the system's performance in terms of the reliability index. They also proceeded in constructing seismic fragility curves as a function of increasing earthquake intensity. Johari et al. (2019) [15] presented a probabilistic framework incorporating a limit equilibrium analysis and a Monte Carlo simulation, by treating the controlling parameters of the stability of a triangular wedge as random variables following a truncated normal distribution. Moreover, Zhou et al. (2021) [16] analyzed the slope stability of a soil–rock mixture, based on a Monte Carlo algorithm, considering the dispersion of strength of soil–rock mixtures. The spatial variability of mechanical properties of geo-materials has been investigated by Liu et al. (2022) [17] as well, by applying the concept of the "Gene" from biology and proposing a promising method to describe the properties of geo-materials based on generic features of specific materials belonging to a certain region. Moreover, machine learning algorithms have been implemented by Zhang et al. (2022) [18] to develop a reliability analysis method to evaluate the time-variant failure probability of a specific reservoir area in China.

A basic tool for the quantitative and qualitative assessment of the vulnerability of geo-structures is the fragility curves, which express the probability that a structure exceeds a specified damage state as a function of the intensity of the triggering phenomenon. Most of the existing research in the evaluation of the fragility of road cut-slopes or embankments focuses on earthquake-triggered failures and soil-type slopes. In that context, empirical fragility curves have been presented [19], as well as, analytical, based on pseudo-static methods [13,14,20], and numerical, based on the finite element [21] and finite difference methods [22]. Due to the frequent occurrence of rainfall-triggered slope failures, the effect of rain infiltration on the safety factor ($F_S$) has been considered in the fragility estimation as a sole or an accompanying failure-triggering mechanism [23–27].

The existing research has mainly focused on the failures of soil slopes and, to a lesser extent, on rock slopes. The uncertainties regarding the variability of the rock mass properties and its discontinuities have created the need to recognize the weakest parts of rock slopes and estimate their engineering characteristics. In the present study, a methodological approach to assess the stability of rock slopes at a regional scale is proposed using a kinematic analysis and a probabilistic limit equilibrium analysis for plane sliding and wedge failure modes. The workflow adopted is described through its implementation along the main road network of the island of Thasos, located in northern Greece. This work is part of a wider project including the whole region of East Macedonia and Thrace,

in Greece, which aims at the multi-hazard risk assessment of basic infrastructures at a regional scale.

## 2. Materials and Methods

The aim of this work is to assess rock slope instabilities at a regional scale. During the last decades, fragility curves have been a useful tool for evaluating the integrity of engineering systems under seismic loads; however, other load types have been considered as well, depending on the utilized engineering model. Hence, to accomplish the goal of the current study, it was deemed appropriate to incorporate rock slope fragility curves. The workflow adopted is shown in Figure 1 and every step is subsequently analyzed.

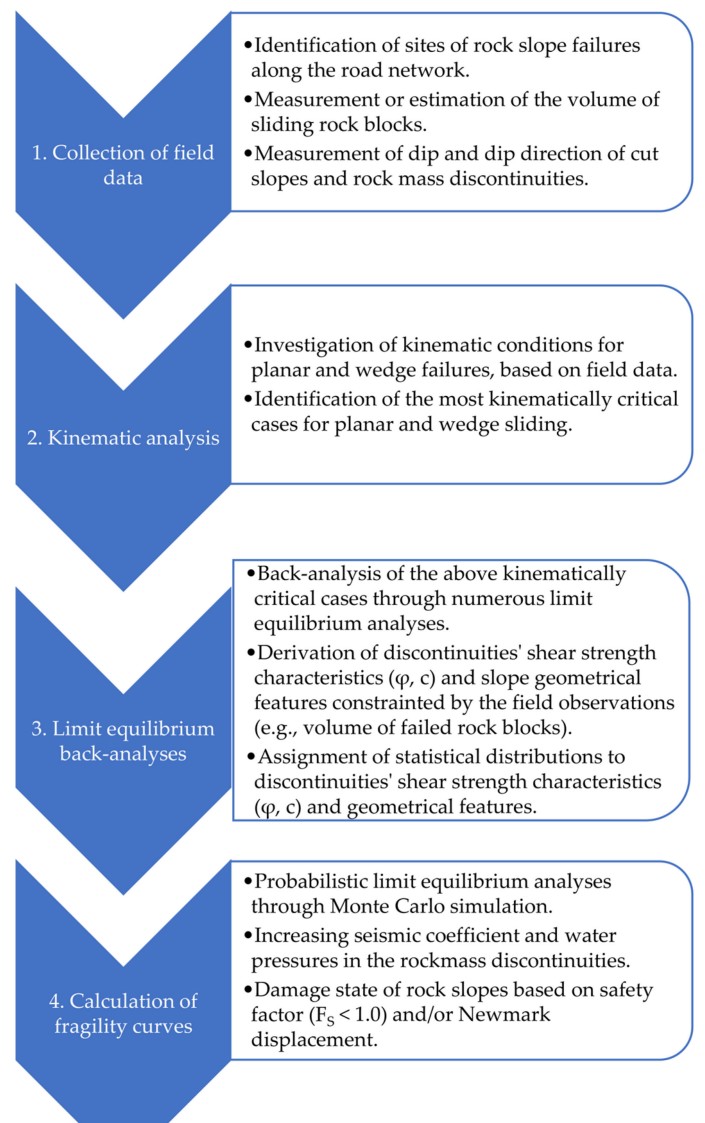

**Figure 1.** General workflow adopted in the present study to derive fragility curves of rock slopes at a regional level.

*2.1. Description of the Study Area*

The general workflow proposed in Figure 1 was implemented on the island of Thasos, located in northern Greece, within the region of East Macedonia and Thrace. The area of the island is about 379 km² and its total population is about 11,520 people. The main road network is developed along the perimeter of the island and it is about 100 km long. The

most populated towns and villages on the island are located along the main road network, whereas smaller road segments link the mountainous villages to the main network. Within this study, the main road network was investigated and is presented in Figure 2, along with the geological map of the island provided by the HSGME (Hellenic Survey of Geology and Mineral Exploration) [28]. The island of Thasos belongs the Rhodope Massif and its geological setting is mainly composed by alternations between marble complexes (mr), gneisses (gn), and schists (sch), which are separated by a transition zone (d). On the perimeter of the island and around areas where water flow occurs, alluvial deposits may be found (al, H.t, and H.sc).

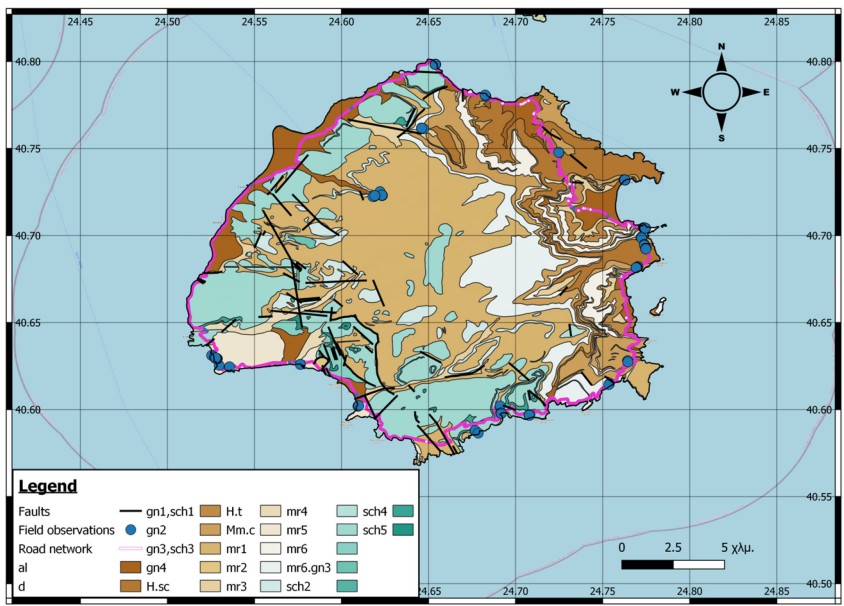

**Figure 2.** Geological map provided by the HSGME and field observations along the main road network on the island of Thasos (background layer is attributed to ©OpenStreetMap).

## 2.2. Description of Workflow Steps

Within the framework of step 1 of the workflow presented in Figure 1, field campaigns took place around the island of Thasos to explore the condition of the cut slopes along the main road network. The outcome of the field campaign included a collection of field data from representative sites where rock slope failures were observed. The field data included photos, measurements of the geometrical features of joints, such as dip and dip direction, as well as estimates of the volume of the rock masses, which were detached from the rock slopes through the mechanisms of wedge or planar sliding. These two mechanisms of rock slope failure were the most common, and they were identified along the road network. Figure 2 presents the representative sites where field data were collected. It should be noted that rock slope failures were observed at other sites as well. However, the sites shown in Figure 2 were deemed the most representative of the failure patterns of rock slopes on the island. Additionally, their location allowed the realization of measurements.

The field observations revealed the presence of steep cut slopes with a nominal inclination of 71° (*v:h* = 3:1). Small to medium rock fractures have occurred at such rock slopes along the road network, with volumes ranging between 0.5 and 2 m$^3$, as shown in Figure 3. Such fractures occurred mostly due to the formation of rock wedges, coming from the intersection of at least three discontinuity planes (Figure 3a,b) and due to planar sliding, where the sliding plane daylights in the slope face and strikes are almost parallel to it (Figure 3b). Since no significant earthquakes have affected the island during the last few years, the observed failures are attributed to water pressure developed within the fissures and tension cracks within the slopes, as a result of rainfalls and/or freeze–thaw cycles of underground water.

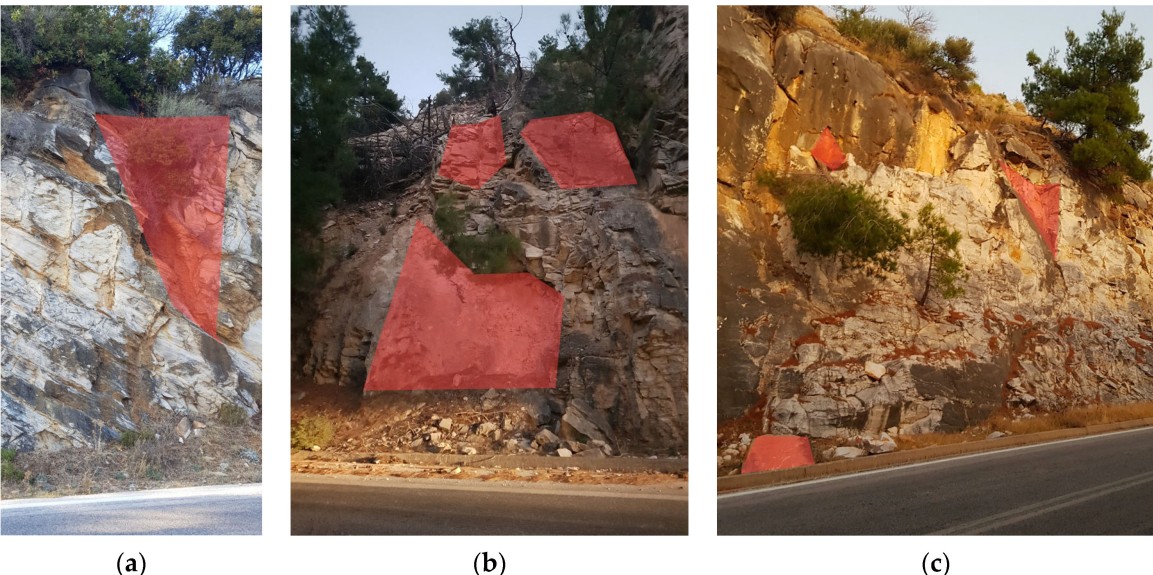

(a)                                        (b)                                        (c)

**Figure 3.** Indicative rock slope failures along the road network of Thasos, due to wedge (**a**,**c**) and planar sliding (**b**).

Within the context of step 2 of the general workflow, the discontinuities' orientation measurements were collectively used to investigate the circumstances under which the kinematic conditions of planar or wedge failures along the road network are satisfied. By means of kinematic analysis, the critical zones, vectors, and intersections were identified on a stereographic projection. The goal of the kinematic analysis was to verify the on-site observations, as well as to highlight the most critical geometrical characteristics of the cut slopes, such as dip and dip direction. The assessment of the kinematic conditions was performed through the software Dips 7.0 of Rocscience (Toronto, Canada) [29].

For step 3 of the general workflow, for the most critical slope face orientations, which were identified through the kinematic analysis for both planar and wedge failures, back-analyses were performed to assess the range of material strength properties of the discontinuities, through limit equilibrium equations [30] (Figure 4). More specifically, for each set of field observations that were identified as critical for either planar or wedge failure, a series of limit equilibrium analyses were performed (~500,000), with cohesion and friction angle of discontinuities randomly sampled from uniform distributions defined within reasonable ranges. Tables 1 and 2 present the range of parameters considered in the back-analyses for plane sliding and wedge failure.

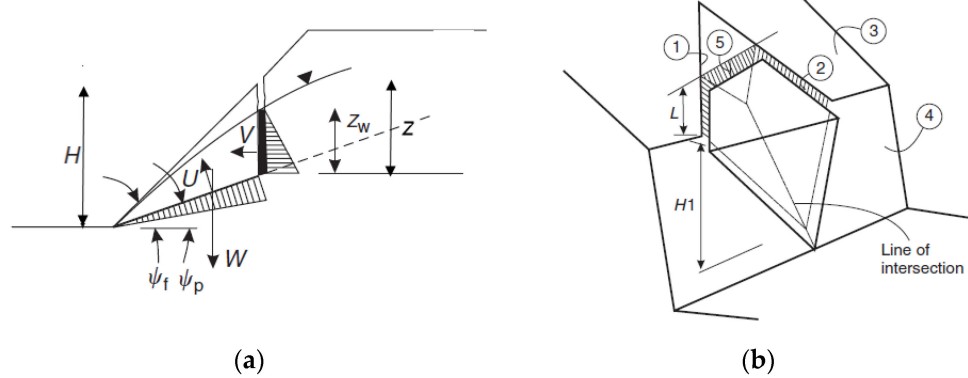

(a)                                        (b)

**Figure 4.** Models of (**a**) plane sliding and (**b**) wedge failure adopted for limit equilibrium analyses (figures from [30]).

**Table 1.** Range of values of the parameters to be determined by limit equilibrium back-analyses for planar failures.

| Parameter | Value |
|---|---|
| Height of sliding mass, H (m) | 1.5–5.0 |
| Dip of slope face, $\psi_f$ (°) | 71 |
| Dip of the upper slope, $\psi_s$ (°) | 20–34 |
| Cohesion of the discontinuity, $c'$ (kPa) | 2–15 |
| Friction angle of the discontinuity, $\varphi'$ (°) | 25–40 |
| Unit weight of the rock mass, $\gamma$ (kN/m$^3$) | 26.0 |
| The ratio of water height in tension crack, $Z_w/Z$ | 0.3–1.0 |

**Table 2.** Range of values of the parameters to be determined by limit equilibrium back-analyses for wedge failures. Indices in dip and dip direction's values correspond to the planes shown in Figure 4b.

| Parameter | Value |
|---|---|
| Height of sliding mass, H (m) | 2.0–3.0 |
| Dip of slope face, $\psi_4$ (°) | 71 |
| Dip direction of slope face, $\alpha_4$ (°) | Based on kinematic analysis |
| Dip of the upper slope, $\psi_3$ (°) | 0–30 |
| Dip direction of upper slope, $\alpha_3$ (°) | 0–150 |
| Dip of tension crack plane, $\psi_5$ (°) | 70–90 |
| Dip direction of tension crack plane, $\alpha_5$ (°) | Same as $a_4$ |
| Distance of tension crack from slope crest, L (m) | 0.5–2.5 |
| Cohesion of the discontinuity planes, $c'$ (kPa) | 2–15 |
| Friction angle of the discontinuity planes, $\varphi'$ (°) | 25–40 |
| Unit weight of the rock mass, $\gamma$ (kN/m$^3$) | 26.0 |
| The ratio of water height in fissures, $Z_w/Z$ | 0.3–1.0 |

It should be noted that for planar failures, the dip of the failure plane, $\psi_p$, was determined by the field observations. Additionally, for wedge failures, the dip and dip direction of intersecting planes of discontinuities were determined from the field observations. Since the limit equilibrium analysis of wedge stability requires very specific geometrical features of the intersecting planes of discontinuities, the dip direction of the slope face was defined based on the kinematic analysis results, taking into account the most critical orientations.

The aim of the back-analyses was to estimate the material strength properties of the discontinuities and the geometrical features of slopes that were not able to be measured. For each case that satisfied the kinematic conditions for planar or wedge failure, 500,000 limit equilibrium analyses were run with randomly sampled system properties, as described above. Then, only the cases where $F_S \sim 1.0$ and the volume of the failed rock mass, $V_r$, measured up to 2 m$^3$ were considered valid for the back-analysis step. The assigned range of values of the randomly sampled variables (Tables 1 and 2) was affected, on the one hand, by field observations and, on the other hand, by the need to have as many as possible valid back-analysis cases out of the total number of analyses. The valid back-analysis cases were collectively treated and resulted in vectors of material strength properties ($c'$, $\varphi'$) of discontinuities and missing geometrical properties of slopes. Appropriate statistical distributions were then fitted to each of these vectors to obtain a more generic description of the variables controlling the problems of rock slope planar and wedge failures. The common practice found in the literature [5,11,14–16] was followed regarding the type of statistical distributions by assigning normal or lognormal distributions.

Having described most of the controlling variables probabilistically, step 4 of the general workflow was implemented. A large number (~30,000) of system realizations was performed, where each of the controlling variables were sampled through the previously defined statistical distributions. Valid realizations were considered those for which $F_S > 1.0$ 214 when no water pressures or seismic loads are applied on the slopes. For each valid realization, the seismic coefficient and/or the percentage of water fill in the fissures

and tension crack gradually increased. The seismic coefficient varied between 0 and 1, with an increment of 0.02, and for each case, $F_S$ was calculated. It should be noted that wedge sliding is an actual three-dimensional mechanism; hence, the direction of the earthquake load plays an important role in the calculation of $F_S$. Since the direction of the seismic load is unknown, it was decided to vary the direction of the seismic load and adopt as critical the one that results in the minimum $F_S$ value. For each loading level, the probability of failure was computed as the ratio of the realizations that exhibited $F_S < 1.0$ over the total number of valid realizations, to derive a set of fragility curves for rock slope instabilities for the island of Thasos.

The factor of safety has been chosen herein to describe the damage state of the rock slopes. However, as stated in Figure 1, another damage index, which is defined for earthquake triggered failures, is the Newmark displacement, which is computed through the Newmark analysis. If $F_S$ on a potentially sliding surface drops below 1.0 at some point during the ground shaking, it does not necessarily imply an important problem, as what really matters is the amplitude of the permanent displacement. The permanent displacement of the rock and soil slopes can be estimated through a method developed by Newmark [6]. The principle of the method is based on the assumption that the sliding mass is a rigid body on a yielding base. The sliding of the mass occurs when the ground acceleration exceeds the base's yield acceleration. The yield acceleration is usually defined as the acceleration at which the factor of safety reaches a value equal to one. The total permanent displacement under a specific ground acceleration time history is computed as a summation of the displacements of the parts of the time history with an acceleration amplitude larger than the yield acceleration.

## 3. Results

This section presents the results of the main steps, described in the general workflow, with reference to the implementation for the island of Thasos.

### 3.1. Kinematic Analysis

The measurements of the dip and dip direction of rock slope discontinuities, which were taken along the main road network of the island, were input into the software Dips 7.0 of Rocscience [29] and plotted on a stereonet projection to conduct kinematic analyses for planar and wedge sliding. A sensitivity analysis was performed to explore the variation of the critical dip vectors, for planar sliding, and critical intersections, for wedge sliding, with respect to the slope dip direction. Figure 5a presents the results of the sensitivity analysis of the kinematic conditions for planar sliding. The highest percentage of the critical dip vectors are observed for slope dip directions between 124° and 148° (10.7%) and 226°–246° (12.5%). For these slope orientations, Figure 5b,c presents the stereonet projection of dip vectors highlighting the critical dip vectors. The critical dip vectors are also presented in Table 3.

Figure 6a presents the results of the sensitivity analysis of kinematic conditions for wedge sliding. The highest percentage of critical plane intersections are observed for slope dip directions of 136° (21.5%) και 230° (21.9%). For these slope orientations, Figure 6b,c presents the stereonet projection of dip vectors, highlighting the critical plane intersections. Table S1 including the critical intersections at those slope dip directions is given in Supplementary Materials.

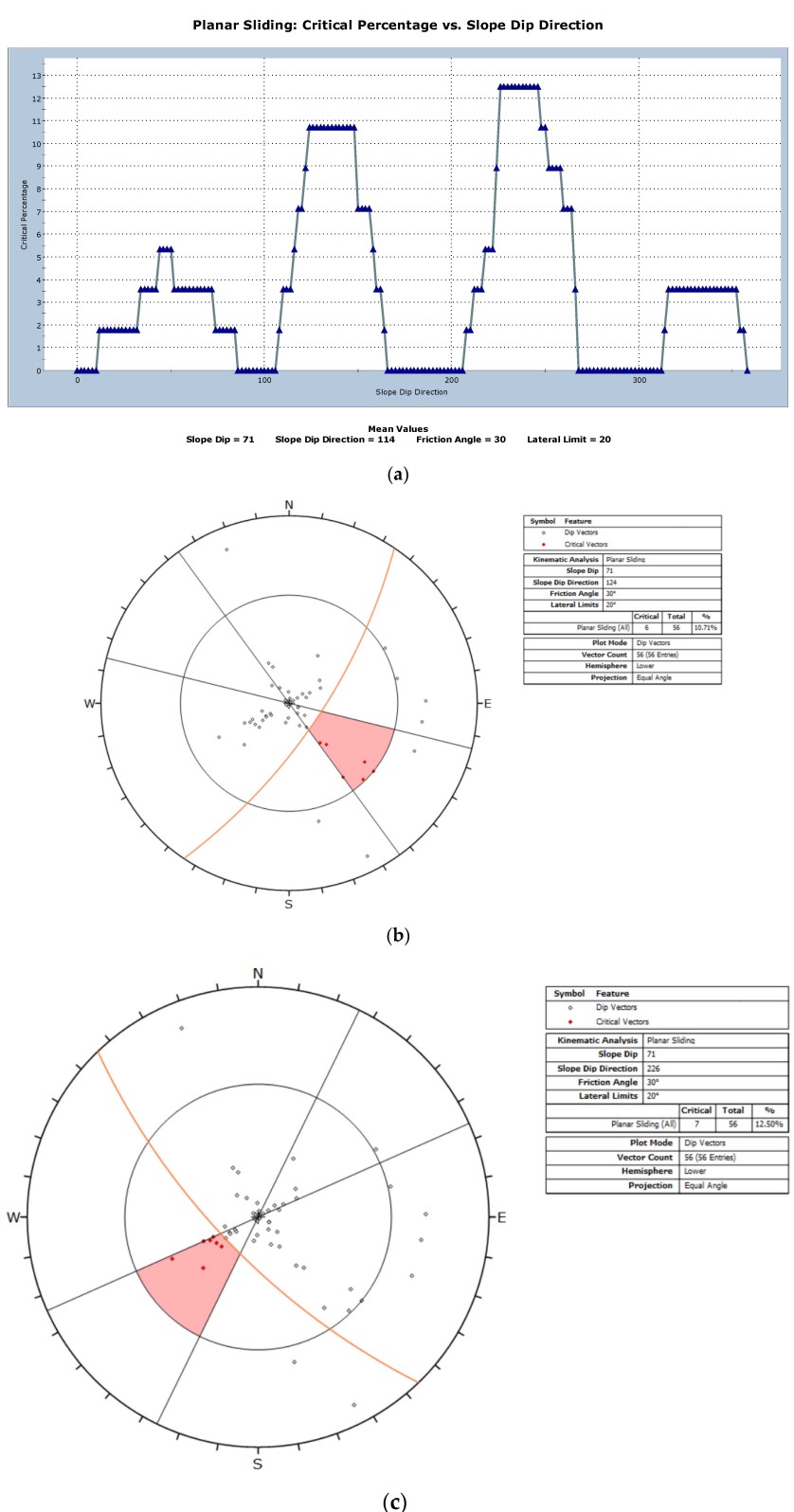

**Figure 5.** (**a**) Results of sensitivity analysis of kinematic conditions for planar sliding, exhibiting the variation of percentage of critical dip vectors with respect to the slope dip direction, and stereonet projection of dip vectors, highlighting the critical dip vectors for slope dip direction (**b**) 124°–144° and (**c**) 226°–246°.

**Table 3.** Critical dip vectors for slope dip direction between 124° and 148° και 226°–246° for the island of Thasos.

| ID | Dip | Dip Direction |
|----|-----|---------------|
| 1  | 36  | 123 |
| 2  | 37  | 138 |
| 3  | 29  | 123 |
| 4  | 31  | 130 |
| 5  | 56  | 132 |
| 6  | 60  | 137 |
| 7  | 66  | 241 |
| 8  | 60  | 240 |
| 9  | 63  | 238 |
| 10 | 45  | 238 |
| 11 | 66  | 232 |
| 12 | 67  | 225 |
| 13 | 54  | 222 |

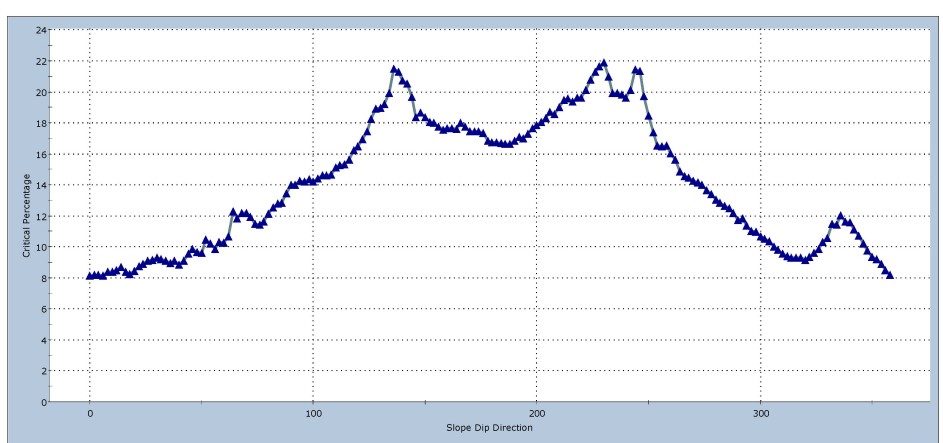

(a)

(b)

**Figure 6.** *Cont.*

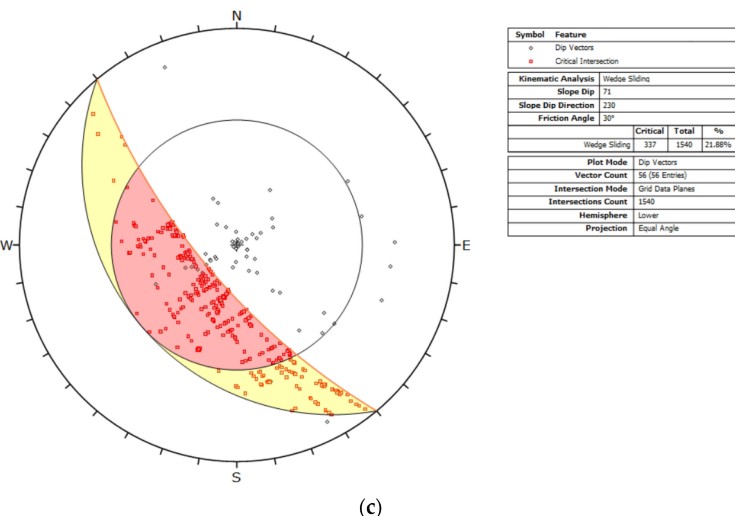

(**c**)

**Figure 6.** (**a**) Results of sensitivity analysis of kinematic conditions for wedge sliding, exhibiting the variation of percentage of critical intersections with respect to the slope dip direction, and stereonet projection of intersections, highlighting the critical ones for slope dip direction of (**b**) 136° and (**c**) 230°.

### 3.2. Back-Analyses for Plane Sliding and Wedge Failure

The critical dip vectors and plane intersections were utilized to perform back-analyses for planar and wedge sliding, according to the procedure described in Section 2.2. Since the dip direction of the slope is not used as an input for the safety factor equation, the critical dip vectors shown in Figure 5b,c were treated together, to obtain estimates of the material strength properties of the discontinuities and missing geometrical features.

The histograms of the material strength properties of the discontinuities and the geometrical features of the rock masses, which were estimated through the back-analysis procedure for the mechanism of planar sliding, are shown in Figure 7.

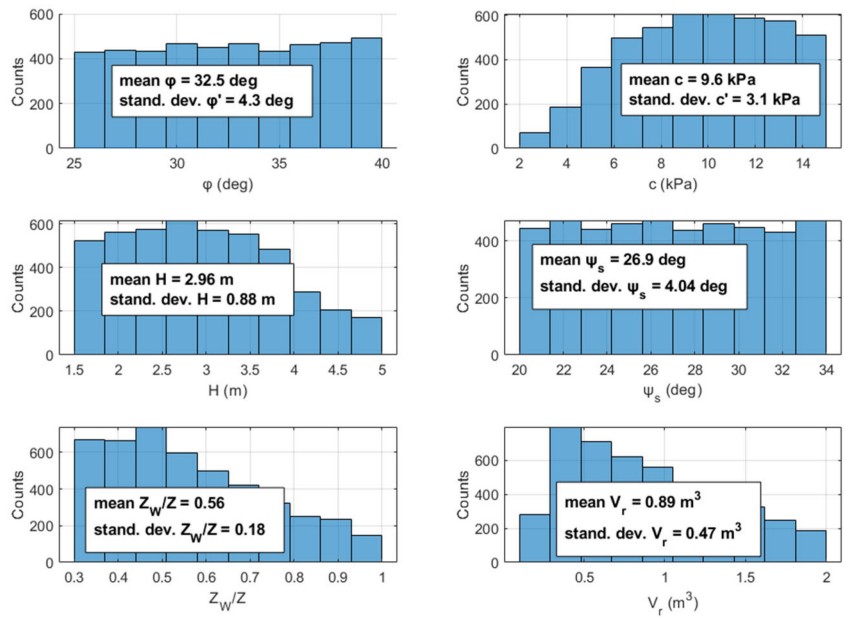

**Figure 7.** Results of back-analysis procedure for the critical dip vectors of planar sliding, in terms of material strength properties of discontinuities and geometrical features of slopes.

The stability analysis of rock wedges is highly dependent on the slope dip direction. Therefore, separate back-analyses were performed for the two slope dip direction ranges where the largest percentage of the critical plane intersections were observed, namely, 128°–140° and 220°–248° (Figure 6a). The histograms of the material strength properties of the discontinuities and the geometrical features of the rock masses, which were estimated through the back-analysis procedure for the mechanism of wedge sliding, are shown in Figures 8 and 9 for the two slope dip direction ranges. In the subplots where bars with two colors are shown, the blue color corresponds to the first discontinuity plane, whereas the red color to the second discontinuity plane.

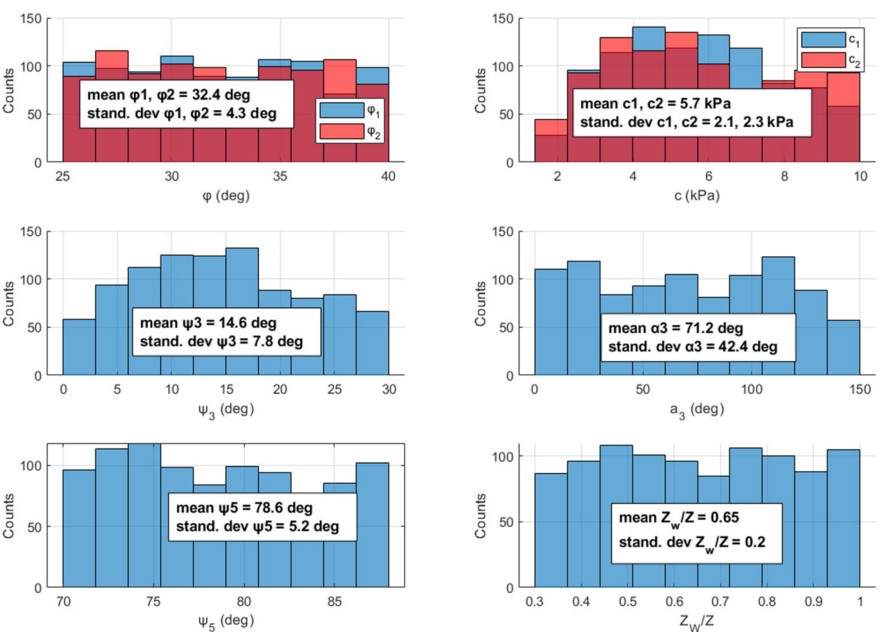

**Figure 8.** Results of back-analysis procedure for the critical plane intersections of wedge sliding, in terms of material strength properties of discontinuities and geometrical features of slopes, for slope dip direction ranging between 128° and 140°.

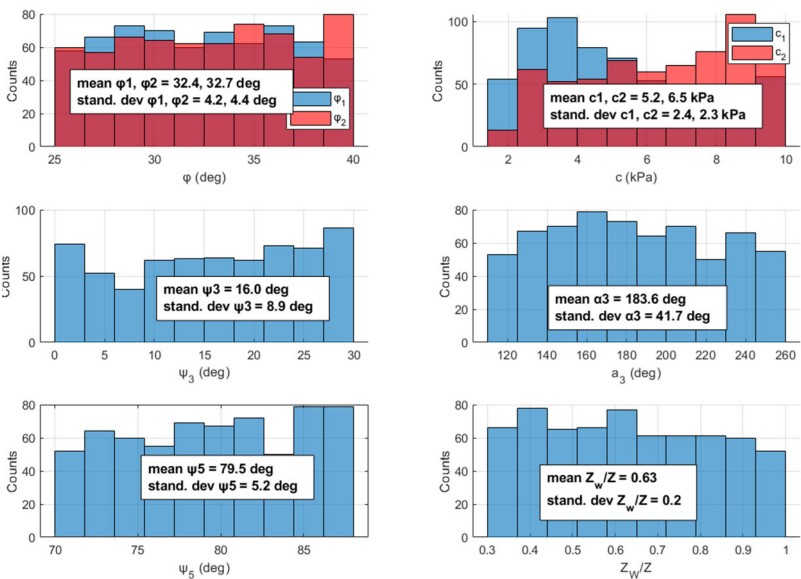

**Figure 9.** Results of back-analysis procedure for the critical plane intersections of wedge sliding, in terms of material strength properties of discontinuities and geometrical features of slopes, for slope dip direction ranging between 220° and 248°.

### 3.3. Monte Carlo Simulation and Fragility Curves

The estimated range of values of the variables controlling the stability of the planar and wedge sliding mechanisms of rock slopes in Thasos formed the basis for the computation of the fragility curves. The back-analyses results were based on field observations that were made at representative sites of rock slope failures on the island. However, the observations' number was limited and they cannot be argued to cover, statistically, the whole road network. Thus, instead of using the resulting statistical distributions of the parameters shown in Figures 7–9, it was decided to define either normal or lognormal distributions for these parameters, using the resulting mean values and standard deviations. Moreover, it was decided to truncate the statistical distributions at the minimum and maximum values of the parameters that came up from the back-analysis procedure. It is the authors' belief that by using the normal or lognormal distributions, the produced fragility curves will have a more general character that will most likely consider cases that were either missed or could not be measured during the field expedition. At the same time, by using the mean values, the standard deviations, and the minimum and maximum values of the parameters that came up from the back-analyses of verified rock slope failures, the produced fragility curves attain a region-specific character.

The Monte Carlo simulation procedure for producing the fragility curves against seismic loading and water pressures has been described in Section 2.2. Figure 10 presents the produced fragility curves for rock slope planar sliding failures, with respect to the peak ground earthquake acceleration ($a_g$). It should be noted that since the scope of the analyses was to assess the stability, rather than to design the slope to be stable, the seismic coefficient, which is actually applied to the sliding mass, was taken as equal to $a_g/g$, where g is the acceleration of gravity.

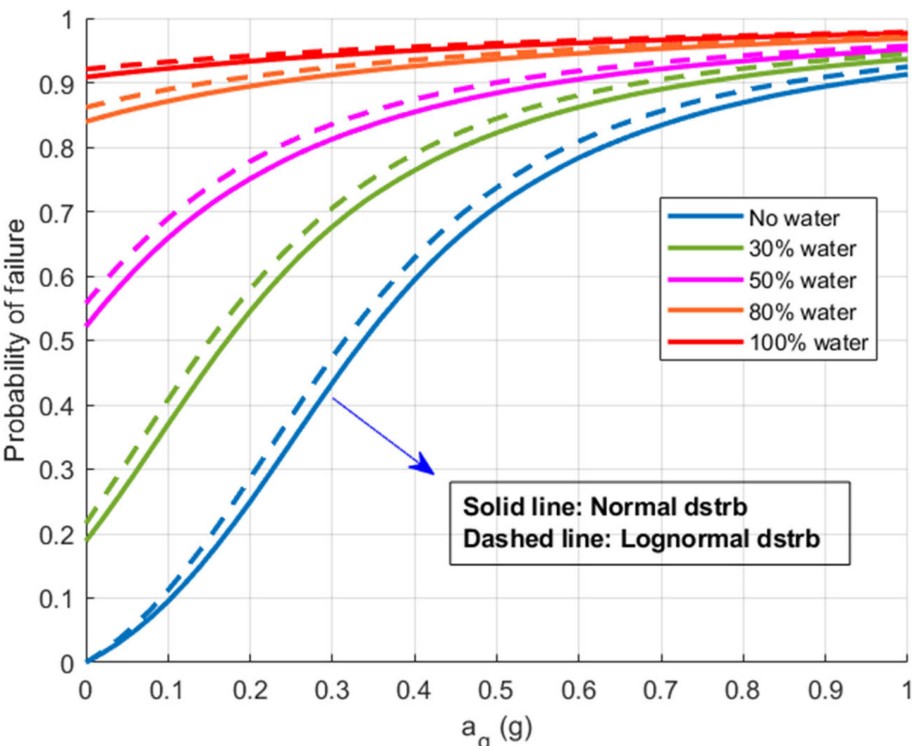

**Figure 10.** Fragility curves of rock slope planar sliding for the island of Thasos, in terms of peak ground acceleration ($a_g$) and percentage of fill of the discontinuities with water.

As denoted by the legend, the various curves, which are plotted in Figure 10, correspond to various percentages of fill of the discontinuities with water. Therefore, the produced fragility curves consider two triggering mechanisms of failure, which are the earthquake and water pressures attributed to either heavy rainfalls or freeze–thaw cycles

of water trapped in the rock mass discontinuities, which are critical for the rock slope stability. It is noted that when no triggering mechanism is present ($a_g$ and water pressures are zero), the probability of failure is equal to zero. However, it should be noted that a zero probability of failure is not equivalent to a satisfactory factor of safety, but it means that the factor of safety is at least equal to one. In this specific case, when no triggering mechanism is present, the mean factor of safety is equal to 1.6 with a standard deviation equal to 0.35. Furthermore, to obtain a probability of failure larger than 50% for dry conditions, the ground acceleration should be as much as 0.35 g. Nevertheless, the development of water pressures at the tension crack decreases this acceleration threshold significantly. Even if 30% of the tension crack height is filled with water, a ground acceleration of around 0.15 g is capable of causing probable planar sliding failures. For larger percentages of water filling, the probability of failure is large enough even without the occurrence of an earthquake. Another observation that can be made from Figure 10 is that assuming either a normal or a lognormal distribution for the parameters controlling the planar sliding problem results in minor differences for the fragility curves, with the latter distribution producing slightly more conservative results.

Figures 11 and 12 present the produced fragility curves for the rock slope wedge sliding failures for slope dip directions 128°–140° and 220°–248°, respectively, as a function of the peak ground earthquake acceleration ($a_g$). The various curves, which are plotted in those figures, correspond to various percentages of fill of the fissures with water. Therefore, the produced fragility curves consider two triggering mechanisms of failure, which are, the earthquake and water pressures attributed to either heavy rainfalls or freeze–thaw cycles of water trapped in rock mass discontinuities, which are critical for the rock slope stability. It is noted that when no triggering mechanism is present ($a_g$ and water pressures are zero), the probability of failure is equal to zero. Nevertheless, as already mentioned in the case of planar sliding, a zero probability of failure is not necessarily equivalent to a satisfactory factor of safety, but it means that the factor of safety is at least equal to one. In this specific case, when no triggering mechanism is present, the mean factor of safety is equal to 6.0 with a standard deviation equal 4.5. Hence, it is observed that wedges formed based on the input variables exhibit a much higher mean factor of safety compared to the planar failure mode, as well as a much higher standard deviation, which could both be attributed to the more complex mechanism of failure if compared to the one of planar sliding. Furthermore, to obtain a probability of failure larger than 50% for dry conditions, the ground acceleration should be as much as 0.54 g. Nevertheless, the development of water pressures at the tension crack decreases this acceleration threshold, but not as much as in the case of planar sliding. If 30% or 50% of the tension crack height is filled with water, the ground acceleration threshold at which wedge failure is expected is not much different from 0.54 g. For 80% and 100% of water filling, $a_g$ around 0.4 g and 0.28 g, respectively, are needed to obtain a probability of failure that is marginally larger than 50%. As observed, the percentage of water fill in the fissures is not as critical for wedge sliding as it was for planar sliding. This may be attributed to the fact that, as explained above, the initial factor of safety of the rock wedges under no triggering phenomenon is significantly higher compared to planar sliding. Another observation that can be made from Figures 11 and 12 is that assuming either a normal or a lognormal distribution for the parameters controlling the planar sliding problem results in minor differences for the fragility curves, with the latter distribution producing slightly more conservative results. However, the fragility curves obtained for the two slope dip directions exhibit differences, mostly in their shape rather than their absolute values, as shown in Figure 13. The largest differences are observed for dry conditions and high $a_g$ values.

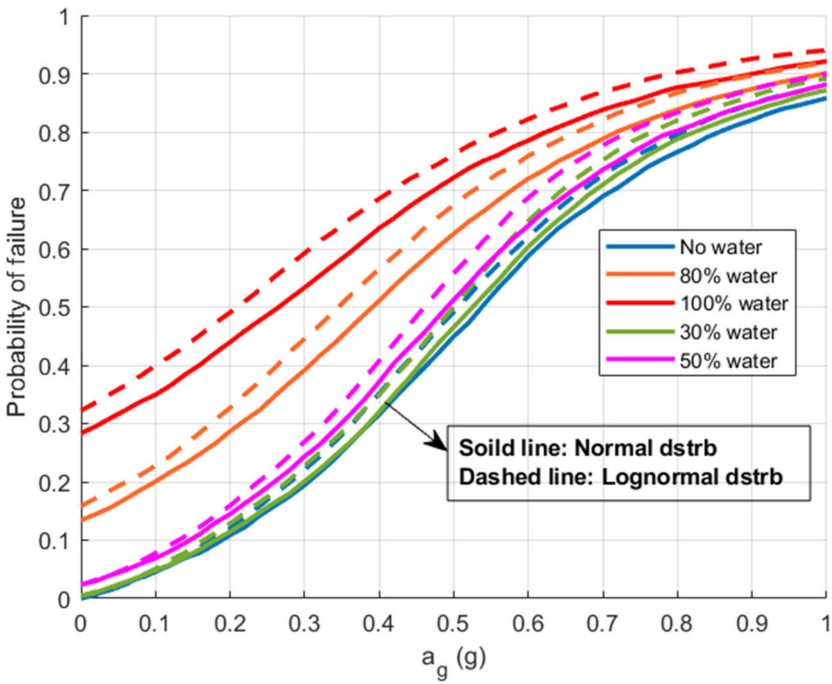

**Figure 11.** Fragility curves of rock slope wedge sliding for the island of Thasos, in terms of peak ground acceleration ($a_g$) and percentage of fill of the discontinuities with water, for slope dip direction between 128° and 140°.

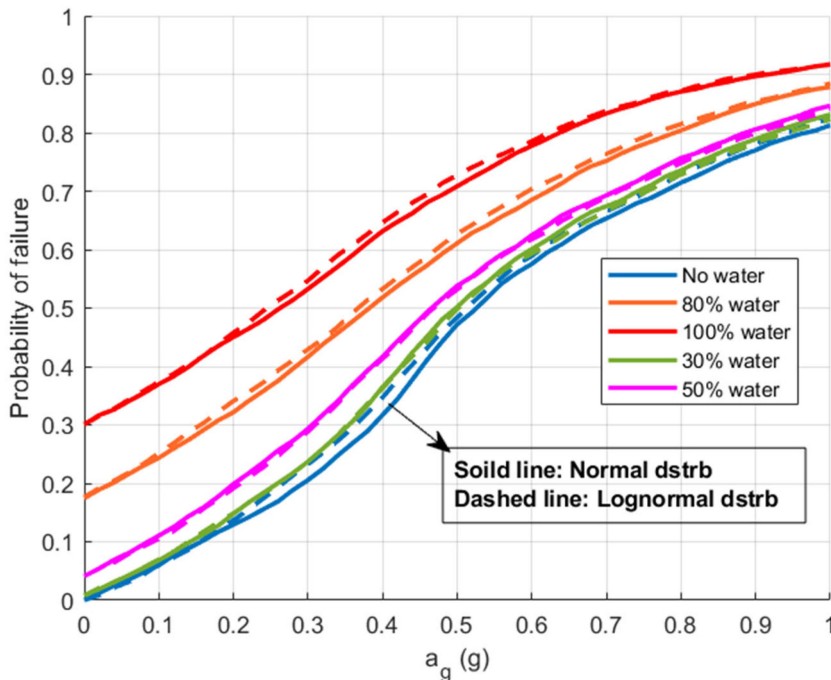

**Figure 12.** Fragility curves of rock slope wedge sliding for the island of Thasos, in terms of peak ground acceleration ($a_g$) and percentage of fill of the discontinuities with water, for slope dip direction between 220° and 248°.

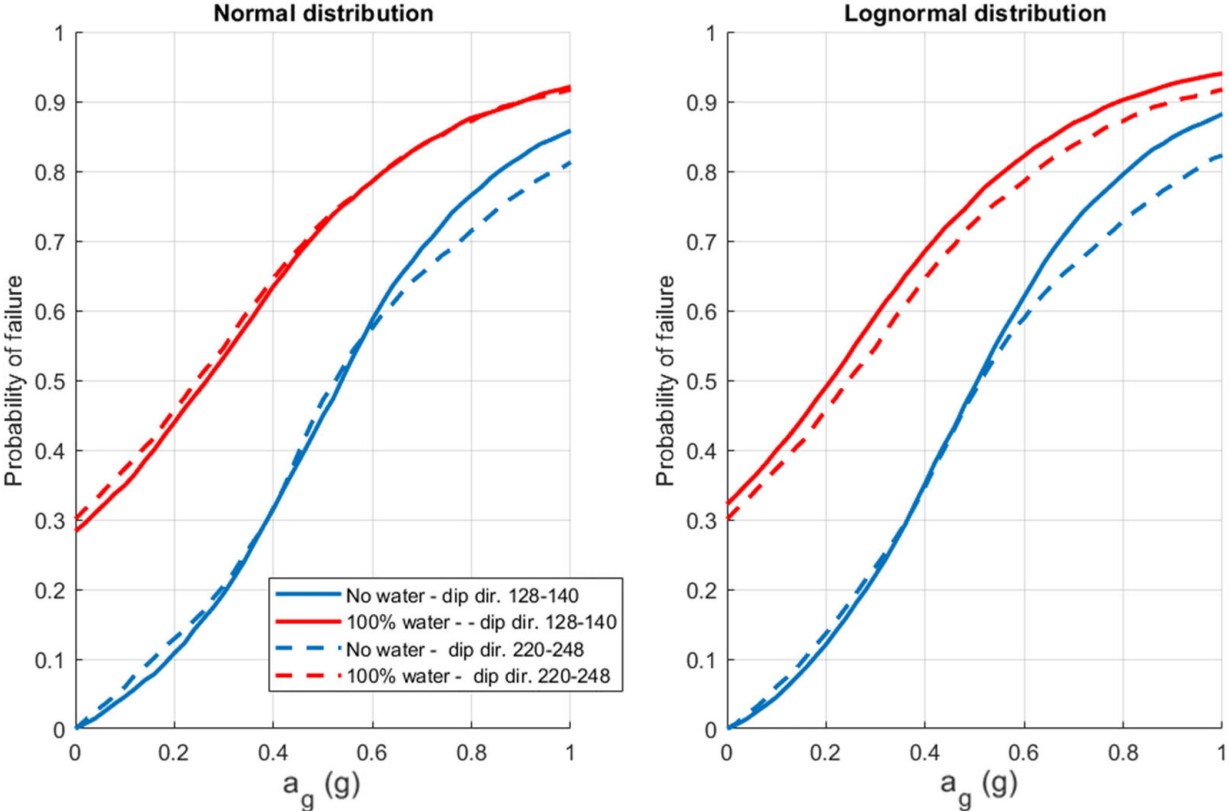

**Figure 13.** Comparison between fragility curves for wedge sliding at different slope dip directions assuming normal distribution (**left**) and lognormal distribution (**right**) for the controlling parameters.

## 4. Discussion and Concluding Remarks

The kinematic analysis results are discussed first. The kinematic analysis for planar and wedge sliding verified the field observations by indicating the critical dip vectors for which the kinematic conditions are satisfied for such rock slope failures. During the field campaign around the main road network of Thasos, a nominal dip of 71° for the cut slopes was observed; therefore, this slope dip value was used in the kinematic analysis. The sensitivity analysis of the percentage of critical dip vectors to the slope dip direction revealed two ranges for the slope dip direction, which include the highest number of critical dip vectors. Surprisingly, for both planar and wedge sliding, the highest percentage of the critical dip vectors and planes' intersections, respectively, was spotted at slope dip directions between 124° and 148° and 226–246°, as shown in Figures 5a and 6a. Figure 14 presents the segments of the road network at which the slope dip direction lies within those critical ranges. These segments are found in the southern, southern-west, and east parts of the island and they are compatible with the field observations. The total length of these segments corresponds to approximately 14% of the total length of the main road network. Such a kinematic analysis, based on field observations, is also useful to assess the susceptibility of the road network to rock slope failures (of planar or wedge sliding type) if the underlying assumptions are satisfied (e.g., if the road segment has an upstream cut slope with a dip angle of around 71°).

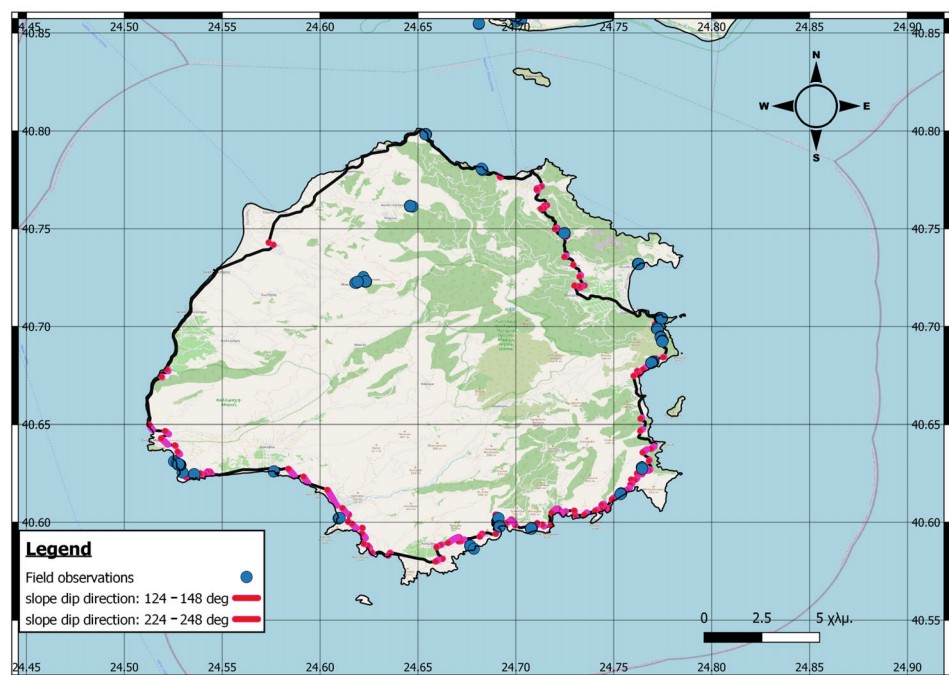

**Figure 14.** Map of the island of Thasos indicating the road network segments that have cut slopes with dip direction within the critical ranges.

The back-analysis procedure included numerous limit equilibrium analyses for planar and wedge sliding for the critical slope dip directions that were identified through the kinematic analysis. For those slope dip directions, the critical dip vectors and intersections were isolated and used as specific cases. The simple, yet theoretically solid, limit equilibrium equations allowed the execution of large numbers of analyses. The known parameters of the problems of planar and wedge sliding were the slope dip and dip direction, as well as, the dip and dip direction of the rock discontinuities and the volumes of detached rock blocks, which were measured during the field campaign. The unknown parameters included the discontinuities' material strength properties, the missing geometrical features (such as the upstream slope dip), and the amount of water pressure within the fissures and the tension cracks, which most likely caused the rock slopes to fail. This procedure, which has been described in more detail in Sections 2 and 3, provided a useful tool to estimate the range of all these parameters, in the absence of site-specific studies, labs, or in situ testing. Furthermore, the collective treatment of the critical dip vectors and plane intersections in the back-analysis procedure was made toward the ultimate goal of this work, which is to assess rock slope instabilities at a regional scale, rather than at a site-specific scale. The resulting distributions of the unknown parameters, shown in Figures 7–9, provided the essential information (mean, standard deviation, and minimum and maximum values) needed to proceed to the last step of the adopted workflow, which is the computation of fragility curves.

The computation of the fragility curves included numerous limit equilibrium analyses of rock slope systems defined in a more generic way than the back-analysis cases. This was made possible through the use of normal or lognormal distributions for all the parameters controlling the stability problems of planar and wedge sliding. The produced fragility curves, shown in Figures 10–12, express the probability of failure of a rock slope in the form of the considered mechanisms as a function of the seismic peak ground acceleration and the percentage of water fill in the rock discontinuities. The correlation of the amplitude of water pressures with natural phenomena, such as heavy rainfalls, is an extremely challenging task, which may demand a site-specific treatment of the problem. However, the produced fragility curves may be useful for the assessment of the seasonal variations of the probability of failure of rock slopes. For example, for the dry months of the island (e.g., summer),

the fragility curves at dry conditions may be more suitable for consideration, whereas the fragility curves that incorporate high water pressures may be more suitable during the autumn or winter months.

The produced fragility curves suggest that plane sliding failure is more likely to happen than wedge sliding, especially in the sole presence of water pressures. However, it should be noted that the slope performance index, which was considered herein, was the safety factor, $F_S$. An $F_S$ value lower than one indicates the occurrence of failure; however, it does not indicate the extent of the failure. More detailed fragility curves could be developed, based on the ones presented herein, by utilizing the concept of the rigid-block displacement analysis [6]. This concept has been incorporated in fragility curves of soft soil slopes [31] by assigning appropriate damage states in terms of the caused disturbance to the road traffic operation.

**Supplementary Materials:** The following supporting information can be downloaded at: https://www.mdpi.com/article/10.3390/geosciences13040098/s1. Table S1: Critical intersections for slope dip direction between 120° and 140° and 20°–248° for the island of Thasos.

**Author Contributions:** Conceptualization, D.S. and N.K.; methodology, D.S and N.K.; software, D.S.; investigation, D.S., S.V., E.I.K. and M.T.; resources, I.D.; data curation, D.S, S.V. and E.P.; writing—original draft preparation, D.S.; writing—review and editing, N.K., S.V., E.P. and V.M.; supervision, N.K.; project administration, I.D. All authors have read and agreed to the published version of the manuscript.

**Funding:** We acknowledge the support of this work by the project "Risk and Resilience Assessment Center–Prefecture of East Macedonia and Thrace-Greece" (MIS 5047293), which is implemented under the Action "Reinforcement of the Research and Innovation Infrastructure", funded by the Operational Program "Competitiveness, Entrepreneurship and Innovation" (NSRF 2014–2020), and co-financed by Greece and the European Union (European Regional Development Fund).

**Data Availability Statement:** Not applicable.

**Acknowledgments:** The authors would like to thank Ioannis Markou, Evangelos Evangelou, and Theofilos Tzevelekis for their valuable help on reviewing and formatting the manuscript.

**Conflicts of Interest:** The authors declare no conflict of interest.

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
