# Peer review of "Toward a Plausible Methodology to Assess Rock Slope Instabilities at a Regional Scale"

_geosciences, doi:10.3390/geosciences13040098_

Round 1

Reviewer 1 Report

The manuscript are understandable and expressed on the basis of almost adequate bibliographic research. Specifically, the authors mainly using kinematic analysis and probabilistic limit equilibrium analysis for plane sliding and wedge failure modes, with a case study in the Thasos location. Some technical issues can be addressed or clarified to improve the quality of the manuscript:

1. The Thasos was taken as a case study, and its detailed geological information was presented. However, these descriptions should be further organized.

2. Figure 1. The authors talk about the damage state of rock slopes based on Newmark displacement. Please explain how Newmark displacement is defined.

3. Figure 4 mentions the limit equilibrium analyses. What method of factor of safety calculation they use (i.e., Janbu's, Sarma's etc. )? Moreover, the information used in the manuscript needs to be indicated in the figure.

4. Table 1 and Table 2. How the range of parameters is obtained? Please provide more information on parameter determination.

5. The relationship between the results obtained and the physics of the phenomenon should be clear. The authors talk about the percentage of water fill in the fissures is not as critical for wedge sliding as it was for planar sliding. Please explain the potential physical mechanism underlying these observations.

6. Please build the link between calculations and case study (e.g., insitu monitoring).

7. Figure 7 need to be modified so that the information in the figure can be clearly distinguished.

8. Recent relevant references missing, such as “D Liu, H Liu, Y Wu, et al. Characterization of geo-material parameters: Gene concept and big data approach in geotechnical engineering. Geosystems and Geoenvironment 1 (2022) 100003”,  “W Zhang, C Wu, L Tang, et al. Efficient time-variant reliability analysis of Bazimen landslide in the Three Gorges Reservoir Area using XGBoost and LightGBM algorithms. Gondwana Research. https://doi.org/10.1016/j.gr.2022.10.004.”

Author Response

We would like to thank the reviewer for his/her time spent on reviewing our manuscript. The reviewer's comments are highly appreciated. In the following, a point-by-point response to the reviewer's comments is provided. 

  1. Text has been added in lines 130 - 132 of the revised manuscript to address the reviewer's comment.
  2. A paragraph has been added in lines 228 - 241 of the revised manuscript to address the comment of the reviewer. Also, another reference has beed added in [29] and the sequence of the references mentioned after this one has been corrected, as well.
  3. The limit equilibrium equations provided by reference [31] were used to calculate the factor of safety for planar and wedge sliding. This is clearly stated in line 175 of the revised manuscript. The values of the parameters used in the manuscript are presented in Tables 1 and 2 for plane and wedge sliding respectively. An additional line has been included in the caption of Table 2 to better correlate the reported parameters with Figure 4.
  4. The determination of the parameters' range is already explained in lines 198 - 208 of the revised manuscript.
  5.  To address the comment of the reviewer, addional text has been included in lines 349 - 350, 369 - 376 and 384 - 386. 
  6.  We would appreciate it if the reviewer could be more specific.
  7.  Figure 7 has been replaced with a better resolution, as suggested by the reviewer.
  8. The references suggested by the reviewer have been included in lines 78 - 84 of the revised manuscript and in the reference list. The numbering of the references has been modified accordingly.

Reviewer 2 Report

Review summary

In the reviewed manuscript, a risk assessment methodology for rock slope failures along the road network of a specific region is presented. The methodology incorporates both field investigations and analytical-probabilistic calculations that results in back-analyzed material and geometrical properties. Then, fragility curves of rock slopes in terms of the most critical triggering mechanisms (i.e. seismic excitation and water presence) are presented that are used to locate the most vulnerable locations of the road network of the examined region.

Review comments

The subject of the manuscript is relevant to the subject matter of the journal and also, to the reviewer’s opinion, it would be interesting for the readers. It is well-written and relies on a coherent background involving rigorous theoretical solutions that also take into account the rock engineering uncertainties associated with rock slopes failure. Further, to the best of the reviewer’s knowledge, it constitutes an original research in an area that generally lacks the necessary attention from the geotechnical engineering community. In order to improve the manuscript the following minor comments are provided to the authors for their consideration.

1.       Line 3: or/end

2.       Lines 108-109: The legend of the figure should be moved under the figure

3.       Line 144: Figure 3 instead of Figure 4

4.       Figure 3: please describe shortly the failure modes shown in (a), (b), (c) of Figure 3

5.       Figure 8 and figure 9: Please correct units for mean and stand. dev. of c1, c2 in subplot of cohesion values

6.       Lines 316-318, “However, it should be noted that a zero probability of failure is not equivalent to a satisfactory factor of safety, but it means that the factor of safety is at least equal to one.”: The reviewer agrees with the quoted comment of the authors. It is noted however that, within the context of a probabilistic analysis for the examined cases of zw/z=0 and ag=0, a zero probability of failure should correspond to a relatively large mean safety factor. For example, by considering the simple “capacity to demand” definition of the safety factor, for uniform capacity and demand random variables with a coefficient of variation of 15% for both capacity and demand, the mean safety factor should approach 2.0. It is recommended that the authors add a comment for the mean factor of safety of their calculation for those cases.

7.       Lines 336-343: the results presented by the authors confirm the general perception that a rock wedge failure is much more difficult to occur than planar sliding, due to the geometrical constraints. The authors are encouraged to comment on this observation.

Recommendation

To the reviewer’s opinion, the manuscript needs minor revisions and can be accepted for publication in the Geosciences Journal. The minor revisions are suggested mainly for some text editing (e.g. correction in figures 8 and 9) and for the above suggestions for the consideration by the authors. Otherwise, the manuscript could be accepted in its present form. The reviewer does not consider it necessary to re-review the revised version of the manuscript.

Author Response

We would like to thank the reviewer for his/her valuable comments on our manuscript. In the following, the response to the reviewer's comments are provided.

  1. The text has been edited according to the reviewer's suggestion.
  2. Figure 2 presents the geological map of Thasos. We believe that the legend of the map should remain within the borders of the map, since it does not overlay any essential information of the map.
  3. The text has been edited according to the reviewer's suggestion. 
  4. The failure modes shown in Figure have been briefly described in lines 154 - 157 of the revised manuscript, according to the reviewer's comment.
  5. We thank the reviewer for noticing this typo. Figures 8 and 9 have been corrected.
  6. We appreciate the comment of the reviewer. Comments regarding the mean safety factor and standard deviation for the case where no triggering mechanism is present have been included in lines 349 - 350 and 369 - 374 of the revised manuscript.
  7. Comments on the observation made by the reviewer have been added in lines 374 - 376 of the revised manuscript.